# Influence of Inter-Firm Network Relationships on Circular Economy Eco-Innovation Adoption

## Shyaam Ramkumar

Department of Social and Political Sciences, University of Milan, 20122 Milan, Italy;
shyaam.ramkumar@unimi.it; Tel.: +39-349-5168243

**Abstract:** Research has shown that inter-firm networks and relationships play a key role in innovation adoption; however, these concepts have not specifically been applied to study their role in the adoption of circular economy eco-innovations. This paper considers whether the embedded relationships within inter-firm networks also influence circular economy eco-innovation adoption. Using a historical case study of the REALCAR closed-loop recycling initiative, by Jaguar Land Rover, from 2013 to 2017, the paper conducted qualitative interviews to reconstruct the structure and nature of the relationships between Jaguar Land Rover and its suppliers. This was complemented with a network regression analysis to determine the influence of these relationships on the adoption and implementation decisions of the closed-loop recycling process by the suppliers of Jaguar Land Rover. The results show that Jaguar Land Rover's relationship as a key customer, facilitation of knowledge sharing among peer suppliers, and resistance from suppliers impacted by changing supply chain relationships played a role in the adoption decisions and adoption timeframe of the REALCAR closed-loop recycling innovation. This has implications for companies and supply chains to consider leveraging the inter-firm relationships embedded in their supply chain networks to accelerate the adoption of circular economy eco-innovations.

**Keywords:** circular economy; closed-loop recycling; eco-innovation; inter-firm networks; embedded relationships; supply chain

---

## 1. Introduction

Reducing the environmental impact and waste of our current global supply chains requires a transformation of how resources are used to produce goods and services. It is necessary to shift supply chains towards a model where resources are recovered and circulated back into the production process. A more circular use of resources has been gaining momentum in recent years as an alternative approach to the current economic model. Defined as the circular economy, it is an economic system that aims to reduce, reuse, recover, and recycle materials in production and consumption processes [1].

Specifically, a circular production approach creates closed-loops to ensure that wasted resources at each step of the supply chain are recovered and recycled [2]. An ideal closed-loop recycling process returns any waste material back into the production process, as close as possible to the source of the waste. It significantly reduces the environmental impact of production by minimising the need for new virgin resources and the treatment of generated waste, while improving profitability [3].

To achieve this, there is a need to better understand how to engage actors within the supply chain towards a common closed-loop strategy [4] and adopt eco-innovations that can make the value chain more circular [2]. Implementing closed-loops effectively and at scale requires the involvement of multiple stakeholders in the supply chain and the alignment of various economic, technical, and environmental factors between these companies [3]. Exploring the network of relationships and

connections between these firms can uncover ways to coordinate the implementation and adoption of innovative closed-loop recycling processes to achieve a more circular supply chain.

Previous research has shown that inter-firm networks play a crucial role in innovation adoption [5]. The embedded relationships and ties within the network of firms, such as hierarchical relations, competitive or cooperative relationships, supplier or economic relationships, and transfers of knowledge and technology [6], enable firms to adapt to changing conditions and make important decisions regarding the implementation of innovations. However, these concepts are used in a limited fashion to understand the diffusion and adoption of eco-innovations [7] that focus on a circular economy, such as closed-loop recycling.

Compared to normal innovations, eco-innovations have higher levels of novelty and uncertainty [8]. As a result, the networks and relationships between firms could be an even more important factor for them to gain knowledge and support in the adoption of eco-innovations [9]. While the literature often focuses on the internal and external drivers for adoption and the management of eco-innovations [7], more research is needed on how the eco-innovation behaviour of networks of firms and the socio-economic structure of interactions between firms and stakeholders across the supply chain influence the effective implementation of circular economy eco-innovations [10,11].

To fill this gap, this paper aims to explore how the embedded relationships within the inter-firm network of companies across a supply chain influence the adoption of closed-loop recycling eco-innovation. The paper uses a historical case study approach to study the network of suppliers of Jaguar Land Rover during the implementation of the REALCAR closed-loop recycling initiative between 2013 and 2017. Through qualitative interviews with managers and employees of Jaguar Land Rover and their suppliers, the paper reconstructs the dynamics and changing structure and nature of the relationships between companies as they adopted REALCAR. Information from the interviews combined with additional data provided by interviewees was used to conduct a network regression analysis to further explore the influence of supplier relationships on the adoption timeframes of REALCAR. By focusing on circular economy eco-innovations, specifically closed-loop recycling, using a mixed-methods approach of qualitative interviews and network analysis, the paper contributes to the existing research on embeddedness, inter-firm networks, and eco-innovation adoption.

The rest of the paper provides a review of relevant literature on the relevance of inter-firm networks and embedded relationships to innovation adoption, as well as the unique characteristics of circular economy eco-innovations that spur the research question. Next, a brief introduction to the Jaguar Land Rover REALCAR closed-loop recycling initiative as an eco-innovation case study is presented. This is followed by a summary of the qualitative research methodology and an overview of the network regression analysis conducted. The findings from the interviews and the results of the network regression are then presented. Finally, the paper ends with a discussion of the results and conclusions.

## 2. Literature Review

Eco-innovations are a key pathway to achieving sustainable production systems, closed-loop supply chains, and product-service systems that lead to a more circular economy. Based on a review of 114 definitions of circular economy, Kirchherr et al. [1] define it as "an economic system that replaces the 'end-of-life' concept with reducing, alternatively reusing, recycling and recovering materials in production/distribution and consumption processes." While eco-innovation is defined as "the production, application or exploitation of a good, service, production process, organisational structure or management or business method that is novel to the firm or user and which results, throughout its life cycle, in a reduction of environmental risk, pollution and the negative impacts of resource use compared to relevant alternatives" [12]. Circular economy eco-innovations such as closed-loop recycling ensure that wasted resources at each step of the supply chain are recovered and recycled back into the production process, as close as possible to the source of the waste, creating resource loops [3]. Such innovations involve entire value chain transformations that employ new

methods of production to ensure resources are recirculated and preserved, so that their value is recovered [2,13].

The processes through which such innovations emerge and spread throughout the different industries are complex, iterative, and evolutionary. Generally, such innovations involve the interaction of a network of organisations which contribute and exchange knowledge and resources to generate, adopt, and disseminate new products and processes [14]. These firms have different contexts, constraints, and incentives to innovate, which are not always aligned with profit-seeking motivations and are influenced by various economic, social, and structural factors. All of these elements result in systems of innovation [15], and understanding the dynamics of such interactions and relationships of the inter-firm networks within these systems is crucial to influence the wider adoption of circular economy eco-innovations.

## 2.1. Inter-Firm Networks and Relationships in Innovation Adoption

Inter-firm networks are defined in the literature as modes of organising and coordinating economic activities between firms. They emerge as a result of firms trying to take advantage of the differentiation and asymmetry of knowledge and resources, creating an interdependence between firms [16]. The types of relationships and links in these inter-firm networks can be economic, through the contractual transaction and interchange of resources, as well as social, resembling the social ties between individuals. For firms, these social ties can be hierarchical relations of authority and power, competitive or cooperative relationships, transfers of knowledge and technology, interpersonal ties between employees and interlocking directorates, or joint membership in associations [6,16].

According to Granovetter [17], these social and behavioural inter-firm ties are embedded in economic relations between firms, explaining economic outcomes. The position and structure of the network of social ties (structural embeddedness) and the characteristics and quality of these relations (relational embeddedness) [5,18] can enable and constrain particular activities and decisions of firms. Institutional theory suggests that the structure and nature of these relationships can impose various coercive pressures due to the dependency on other organisations, mimetic pressures to imitate more successful organisations for legitimacy, and normative pressures to change their goals or develop new practices [19]. Moreover, these ties are multiplex and can exist simultaneously and shape firm behaviour in different ways [6,20].

Research has shown how inter-firm networks and embedded social relations are critical to the adoption of innovation. Ozman [5] states that innovation is a collective and evolving process in which the networks and relationships between firms play a key role. Inter-firm networks allow companies to access necessary resources, as well as to learn from, and imitate, other firms, particularly ones they trust and are socially connected to. Embedded relations in inter-firm networks influence the propensity of firms to "innovate, take risks, and act proactively" [21]. Like individuals, firms use their interactions with other firms within their network to make sense of others' behaviours and to make decisions. The structure of inter-firm networks, inter-firm dependence, and the strength, frequency, and quality of ties with other firms are important factors that determine firm-level entrepreneurial behaviour [21].

For example, Dhanaraj and Parkhe [22] point out how hub firms, that are centrally located in their network and have a certain level of power, can act in a leadership capacity to orchestrate innovation. These firms utilise their network position and relations to bring together resources from their network members, share knowledge to parts of the network where it is needed, and manage the relationships to enhance socialisation within the network. In this way, hub firms foster the adoption of innovation by making use of the structure and nature of the formal and informal relationships within their inter-firm network.

Research by Öberg [23] focuses on six case studies of companies that adopted incremental, radical, or disruptive innovations and the characteristics of their business networks. The paper found that incremental innovations that create improvements utilise existing networks and strong social ties. On the other hand, radical innovations that bring new ideas to the market are brought about by a focal

party utilising weak ties and changing roles of current business partners. Lastly, disruptive innovations that challenge the existing structure are brought about by strengthening ties with new entrants and weakening ties with current partners. Moreover, these innovations, in turn, can affect the structure and nature of the relationships in the network.

Robertson, Swan, and Newell [24] explore the adoption and diffusion of innovation in computer-aided production management (CAPM) technology using a case-study approach of three companies. Using theories of institutional isomorphism [19], their research explored whether the adoption of CAPM technology is influenced by coercion from other organisations the companies are dependent on, imitation of peer firms that have successfully adopted the technology, and industry norms that pressure them to adopt in order to seem legitimate. Their analysis found that the adoption of CAPM innovation was influenced by embedded inter-firm network relationships, such as suppliers that pushed the technology, informal contacts with other firms which implemented the technology, and professional associations that firms were connected to.

### 2.2. Relevance for Eco-Innovation Adoption

The factors that influence innovation adoption are also important for eco-innovations; however, they "will probably not influence the same variables with the same strength" [25]. Compared to normal innovations, eco-innovations have higher levels of novelty and uncertainty [8], since companies tend to bear higher costs in order to create greater societal benefits, which puts them at a disadvantage relative to their polluting competitors [26]. Therefore, companies require greater internal capabilities and resources to adopt eco-innovations [8,25,27,28]. In addition, companies also face greater external pressures from regulatory factors and market demand from customers to overcome the low incentives to adopt eco-innovations [9,25,28].

The existing research on eco-innovations mentions the importance of stakeholders and inter-firm networks for awareness and adoption. Eco-innovations require greater external knowledge and cooperation with partners than traditional innovations [26]. Due to the higher uncertainty of eco-innovations, it is even more important for firms to cooperate with external partners, suppliers, and other stakeholders within their network to share knowledge, gain financial support, and mitigate the risks and costs of adopting novel eco-innovations [8,9,28–30]. Inter-firm networks can influence the internal and external decision criteria for eco-innovation adoption [28], as suppliers and business partners within a firm's network encourage firms to be more aware of, and adopt, a pro-environmental behaviour [27,31,32].

Additional research utilising institutional and stakeholder theory suggests that there is evidence of coercive, mimetic, and normative pressures within a firm's network on eco-innovation adoption. Firms are influenced by pressures from regulations and customer demands, motivations to stay ahead of their competitors, co-operation with their peers, and industry standards, all of which can dictate whether they adopt eco-innovations [31,33–35]. The structure and nature of the network of relationships and connections to other stakeholders in which these pressures are embedded play a significant role in eco-innovation adoption [36,37].

However, there is no explicit research focusing on how embedded relationships in inter-firm networks influence circular economy eco-innovation adoption. More theoretical and empirical approaches for evaluating eco-innovation behaviours by networks of firms and relevant stakeholders are needed [10,38]. A deeper understanding of how the socio-economic structure of interactions within a network of businesses and other stakeholders could influence the effective adoption and implementation of circular economy material flows lacks a strong link and acceptance among researchers [11]. The relevance of theories regarding the mechanisms through which such eco-innovations get adopted and diffused through social norms and social networks is not yet established [7].

Circular economy eco-innovations such as closed-loop recycling are sensitive to interactions and inter-related developments between businesses, society, and institutions due to their systemic nature [2,10]. Such eco-innovations must build on, and modify, existing management and production

structures, coordination processes, and social aspects to be successful [13]. Therefore, it is important to understand how the structure and nature of the relationships within inter-firm networks influence the decision to adopt circular economy eco-innovations.

This paper aims to address this area of research on eco-innovation adoption and circular economy. The research intends to utilise previous evidence and theories on the role of embedded relationships within inter-firm networks on the adoption of innovations and eco-innovations to study the adoption of closed-loop recycling. The objective is to determine the importance of these factors given the differences in characteristics of eco-innovations compared to traditional innovations.

## 3. Materials and Methods

### 3.1. Case Study of Jaguar Land Rover's REALCAR

In order to explore the role of embedded relationships in inter-firm networks on circular economy eco-innovation adoption, this paper utilises a case study approach. As mentioned by Uzzi [39], this approach provides rich data to conduct a detailed analysis of inter-firm ties and their dynamics. Though the approach has "moderate generalizability" [39], the strength of the case study approach is that it fills gaps of knowledge present in more quantitative statistical and modelling methods [40]. Moreover, it enables the study of concepts and indicators that have no quantitative measure, such as power dynamics, culture, trust, and other factors embedded in relationships and networks.

Specifically, the paper studies the case of the Jaguar Land Rover REALCAR closed-loop recycling project. The REALCAR project was an initiative by Jaguar Land Rover and a variety of stakeholders and partners that came together to develop a process to collect, recycle, and reuse aluminium waste material in the production of automobile bodies. When it was developed and implemented, REALCAR was one of the first best-practice examples of a successful, large-scale implementation of closed-loop recycling within the automotive supply chain. Moreover, it paved the way for other automakers such as Ford, BMW, and Audi to implement similar initiatives later on.

The history of the REALCAR project can be traced back to 2002, when Jaguar Land Rover decided to produce automobiles from aluminium instead of steel, the traditional material used by automakers. Aluminium was chosen to "reduce weight, improve fuel consumption and tailpipe emissions and reduce costs to the user" [41]. Since aluminium is more expensive and energy-intensive than steel, Jaguar Land Rover started looking for ways to reduce these costs and impacts, particularly as Jaguar Land Rover shifted more of their production towards aluminium [42].

Around 2007, after realising that a key way to achieve this goal was to utilise recycled aluminium at every stage of production, Jaguar Land Rover developed the REALCAR project. Receiving 1.3 million British pounds in funding from a collaborative R&D grant by the UK government's Innovate UK program, Jaguar Land Rover brought together a consortium of supply chain partners—aluminium producer Novelis, technology consultant Innoval, body stamping supplier Stadco, Brunel University, and others. Together, they researched and developed a new type of aluminium alloy, RC5754, that could utilise recycled waste aluminium material collected from the production process of car bodies without sacrificing performance. This enabled Jaguar Land Rover to recycle the nearly 50% of aluminium waste from the production of their automobiles and recover 90–95% of the value of the material, significantly reducing costs by millions of pounds [43].

However, to fully achieve the benefits of this research and reach their target of using 75% recycled aluminium in their cars by 2020, Jaguar Land Rover needed to move beyond the R&D phase and implement the REALCAR approach across their supply chain. This included implementing the approach in Jaguar Land Rover's own internal production facilities, but more importantly in the production facilities of its external suppliers. REALCAR's implementation required investments in new equipment as well as modifications of existing processes, with Jaguar Land Rover investing more than £7 million across their three facilities, Novelis investing £6 million in their Latchford recycling plant, and nine other external suppliers also making investments and changes to their operations [42].

With these investments, Jaguar Land Rover's internal stamping facilities and external stamping suppliers were able to separate and collect waste aluminium material from the production of automobile car bodies. The new equipment and processes aimed to minimise the contamination of the aluminium waste with steel and other metals. Once collected, the scrap was then baled and transported by scrap dealers, which sent the material by truck to Novelis's Latchford recycling plant. Here, the waste material was re-melted and recycled to produce new aluminium sheets, which were then provided to the stamping facilities to produce new car parts [42]. Figure 1, below, provides an overview of this process, as well as an image showing the implementation for Ford, which happened much later than the Jaguar Land Rover REALCAR initiative but follows a similar approach.

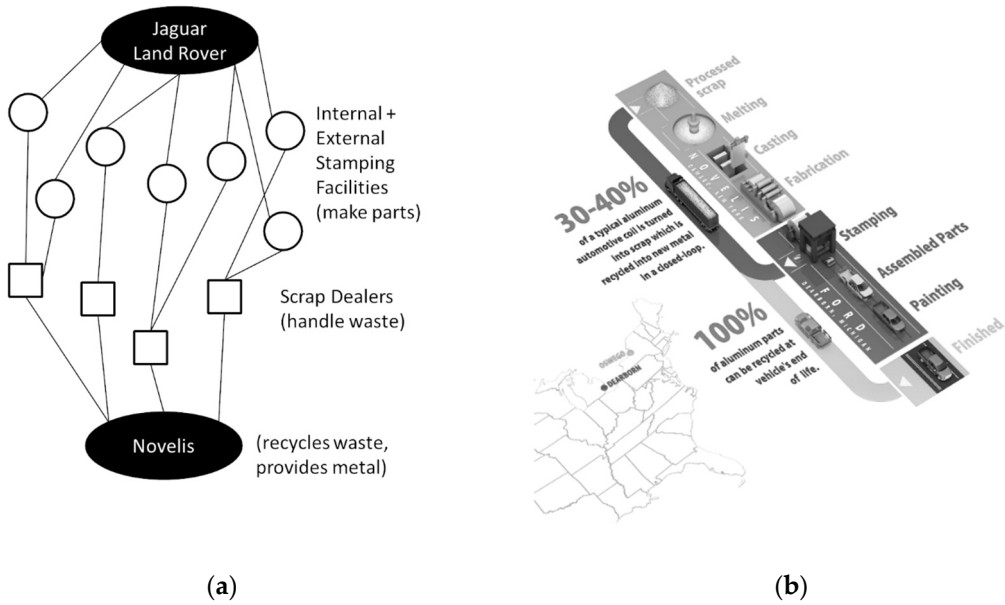

(**a**)                                        (**b**)

**Figure 1.** A conceptual overview of the REALCAR supply chain. (**a**) Supply chain diagram of the stakeholders involved in the REALCAR closed-loop recycling process and their roles; (**b**) Process diagram of REALCAR implementation at Ford [44].

Previous research on the REALCAR closed-loop recycling process has focused on the environmental benefits, technical innovation, and financial investments [45]. In addition to these aspects, there was also a need to coordinate the supply chain network, since REALCAR transformed the value flow within the supply chain network and not all companies benefitted equally. Jaguar Land Rover and Novelis needed to consider the incentives and opportunities for the whole value chain and manage their supplier relationships to effectively engage them to adopt the REALCAR approach [41]. Therefore, REALCAR is an ideal case to understand the role that relationships between firms in the supply chain network play in the adoption of a novel innovative process for closed-loop recycling.

### 3.2. Qualitative Study

To better understand the role of embedded relationships and inter-firm networks in the adoption of the REALCAR closed-loop recycling approach by the suppliers of Jaguar Land Rover, a qualitative research method was first followed. Similar to the approach taken by Robertson, Swan, and Newell [24], the interviews were intended to explore the nature of the relationships and network ties among the different actors and the role they played in the adoption of REALCAR. Through these interviews, the paper gained a description of the sequence of events that led to the adoption of REALCAR and identified emerging patterns that might fit with the existing theory on embedded relationships in inter-firm networks and innovation.

The interviews consisted of a series of semi-structured questions and were conducted for approximately 1–1.5 h over the phone. The topics covered during the interview were based on the types of embedded relationships discussed in the literature, such as hierarchical relations of authority and power, competitive or cooperative relationships, transfers of knowledge and technology [6,16], and on the structure and quality of the relationships [5,18]. Interviewees were asked about:

- Timelines for the adoption of the REALCAR approach
- Motivations for the adoption and implementation of REALCAR
- Nature of relationship between Jaguar Land Rover and internal and external stamping facilities
- Nature of relationship, knowledge sharing between internal and external stamping facilities
- Nature of relationship with scrap dealers
- Ease of implementation in terms of costs, logistics, technical capabilities, contracts, etc.

The interviews were conducted with executives, managers, and employees who were involved in the implementation phase of REALCAR from 2013 to 2017. The interviewees were identified through the snowball technique as outlined by Borgatti and Li [6] in the context of a supply chain. Initial interviews with key decision-makers from Jaguar Land Rover who led the implementation of REALCAR revealed contacts within Jaguar Land Rover, Novelis, internal stamping facilities, external stamping supplier facilities, and scrap dealers. These contacts were then interviewed and asked to provide additional contacts at the various firms.

In total, 61 individuals within Jaguar Land Rover, Novelis, 3 internal stamping facilities, 12 external stamping supplier facilities owned by 7 external stamping suppliers, and 5 scrap dealers were identified. However, due to difficulties in interviewing many of the individuals, who had since transferred from their roles, retired, whose contact information was outdated, or who refused to be interviewed due to confidentiality issues, only 17 interviews were conducted. A summary of the number of individuals identified and interviewed by company type is provided in Table 1.

**Table 1.** Summary of interviewees identified and interviewed.

| Company Category | Number of Companies/Facilities | Number of Individuals Identified | Number of Individuals Interviewed |
|---|---|---|---|
| Jaguar Land Rover | 1 | 19 | 6 |
| Novelis | 1 | 5 | 2 |
| Internal Stamping Facilities | 3 | 4 | 2 |
| External Stamping Supplier Facilities | 12 facilities 7 suppliers | 23 | 2 |
| Scrap dealers | 5 | 10 | 1 |
| **Total** | **22** | **61** | **17** |

Though the number of interviews conducted was lower than the number of individuals identified, the perspectives of the different types of firms involved in the REALCAR project were sufficiently covered. Moreover, relevant information regarding the inter-firm relationships and network structure were provided by key decision-makers interviewed from Jaguar Land Rover and Novelis, who had oversight and contacts with many of the stakeholders involved in REALCAR.

Interviewees were informed in writing of the purpose of the study, and letters of consent were provided for the interviewees to sign. These documents stated that their responses and the information provided would be treated confidentially and any details which would reveal the identity of the individuals interviewed would not be mentioned or disseminated in the research. In addition, following the approach by Öberg [23], the interviews were supplemented with newspaper items, press releases, and email exchanges with some of the interviewees, to get additional information on the relationships and network ties between firms, as well as to triangulate information received from the different sources.

Similar to the approach taken by Uzzi [39], the information gathered from the interviews was interpreted based on expectations and theories derived from the literature review. A coding framework was developed to identify three categories of embedded relationships. First, hierarchical relations of authority and power and coercive pressures were coded to identify whether Jaguar Land Rover used their network position and power as customers to encourage the stamping facilities to adopt REALCAR. Second, mimetic pressures were coded to identify whether competitive and cooperative relationships, as well as interlocks between the stamping facilities, encouraged knowledge sharing, learning, and imitation of best practices that influenced the adoption of REALCAR. Third, changes in the structure and nature of the relationships between the different firms, such as changes to existing contracts or changes to the roles of the companies, were coded to identify their effects on the adoption of REALCAR. The interview notes and transcriptions were analysed using a qualitative data analysis software, QDA Miner, using this coding framework. In the process of analysing the interviews conducted, the framework was modified and refined based on the literature and the information from the interviews.

### 3.3. Network Regression Analysis

Unfortunately, few of the scrap dealers were able to be interviewed due to confidentiality issues and non-disclosure agreements. However, during the interviews, some interviewees provided supplementary historical data related to REALCAR, such as the adoption timeframes of the different stamping facilities from 2013 to 2017, information on which scrap dealers were contracted to handle the waste from each stamping facility at the time, and additional characteristics of the stamping company facilities. Follow-up exchanges with interviewees also uncovered the scrap dealers with which they had difficult negotiations due to resistance to the REALCAR closed-loop approach. Further analysis of this data seemed necessary to better understand the influence of scrap dealer relationships on the adoption of REALCAR.

From the literature, researchers note the potential for network analysis to complement a qualitative case study approach and gain a deeper understanding of embedded network relationships [46]. Edwards [47] points out the benefits of combining qualitative research and network analysis through a literature review of different studies which employed mixed-method approaches. These studies describe how information gathered from ethnographic observations or semi-structured interviews can be quantified into relational network data. Such a mixed-method approach enables the exploration not only of the structure and form but also of the content and processes of network relationships, enabling triangulation to create a narrative that offers greater context.

Consequently, this paper also chose to utilise the information from the interviews and the supplemental data provided by interviewees to conduct a network regression analysis using the Multiple Regression Quadratic Assignment Procedure (MR-QAP) technique, to complement the qualitative interviews. Rather than conducting regressions of dependent and independent variables as in traditional statistics, the MR-QAP method performs regressions of the dyadic ties or relationships between two actors within a network [48]. This network regression method is superior to traditional Ordinary Least Squares (OLS) techniques for dyadic relationship data, as it removes any biases from structural autocorrelation [49].

The network regression sought to understand if two stamping suppliers had the same scrap dealer or if they both faced resistance from their scrap dealers, then how similar was their time-to-adoption in months for the REALCAR approach to be implemented. The unit of analysis for the network regression were the 3 internal Jaguar Land Rover and 12 external supplier stamping facilities—a total of 15 facilities. The ties between these nodes were the dependent network variable, the difference in time-to-adoption of REALCAR between stamping facilities, and the independent network variables, common scrap dealers among stamping facilities, and resistance from the scrap dealers. Figure 2 illustrates the dependent and independent network variables.

**Dependent Network Variable:**
Difference in time-to-adoption between stamping facilities

**Independent Network Variables:**
Stamping facilities with a common scrap dealer

Stamping facilities who faced resistance from scrap dealer

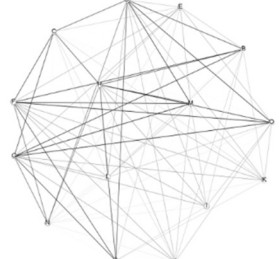
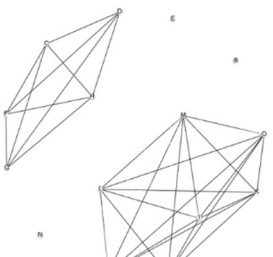
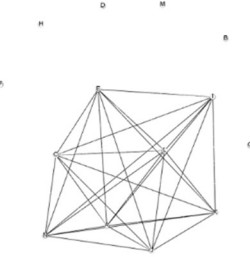

**Figure 2.** Network diagram illustrating the Multiple Regression Quadratic Assignment Procedure (MR-QAP) network regression to test whether the structure and nature of the stamping facilities' relationships with scrap dealers influenced the time-to-adoption of REALCAR. The nodes are the 15 internal and external stamping facilities, while the ties represent the dependent and independent network variables.

To more accurately test the influence of scrap dealer relationships, a network regression model was developed to include other control factors that could influence adoption timeframes. The model followed the approach outlined in Hollenstein and Woerter [50], which used a regression model to identify the influence of different firm characteristics on the adoption and diffusion of technology. However, since we are considering a circular economy eco-innovation, factors relevant to circular economy eco-innovation adoption described in the literature were considered—firm size, firm learning, technological implementation, and financial costs, as well as leadership and governance [25,50,51].

Firm size, measured in the literature by the number of employees or turnover [50,51], was not available at the stamping facility level, and therefore the difference in the size of scrap generated was used as a proxy for firm size. The difference in the order of implementation of REALCAR was used to capture firm learning of best practices by later adopters. Whether two stamping facilities faced significant capital investment or process changes was used to capture implementation factors. The difference in logistics costs was used as a proxy for financial costs. Lastly, whether two stamping facilities were owned by the same company acted as a proxy to capture governance and leadership factors. Figure 3 outlines the network regression model and the network variables used.

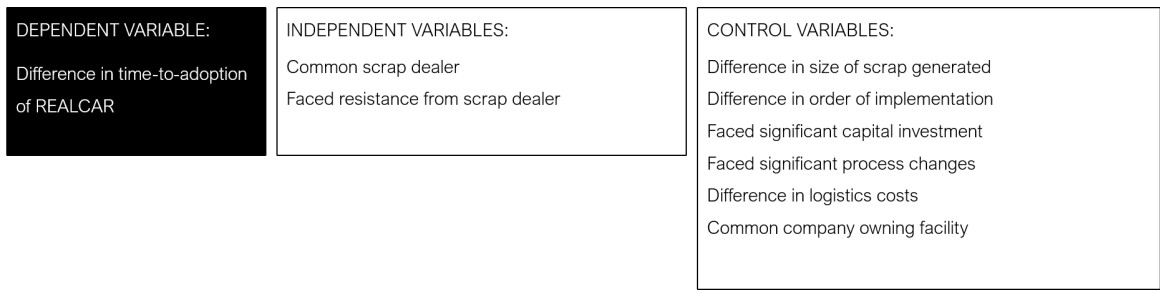

**Figure 3.** Overview of the network regression model and the network variables.

To gather data for these network variables, an Excel database was first created, where each row represented a stamping facility and each column represented the information needed to compute the network regression variables. This Excel database was first populated using the supplementary data tables, provided through follow-up email exchanges with some of the interviewees, and captured the following information:

- Date of first discussions with the stamping facility and date of REALCAR implementation (the difference between these dates was the facility's time-to-adoption in months)

- Scrap dealer contracted to the stamping facility
- Size of scrap generated per month by the stamping facility, in tonnes
- Order of the stamping facility's implementation of REALCAR, from 1 to 15
- Logistics costs for the stamping facility, in GBP per tonne
- The company which owned the stamping facility

Next, information on the resistance from the scrap dealers, whether the stamping facility faced significant capital investment, and whether the stamping facility faced significant process changes was derived from the interviews. A similar approach to that of Mckether et al. [52] was used to convert interview data into social network data. The interviews were transcribed using the software QDA Miner and coded for references to scrap dealer resistance, significant capital investment, and significant process changes for each of the 3 internal and 12 external supplier stamping facilities. If there was any mention of resistance from scrap dealers, capital investment, or process changes for a stamping facility in the interviews, the value in the appropriate column in the Excel database was 1; otherwise it was 0.

As in the approach followed by Coviello [46], this network database was shared with interviewees who provided the data and who had sufficient knowledge of all the relationships within the supply chain network. The data was revised as necessary until the information captured was deemed accurate. This ensured that the data was credible and valid and increased confidence in the network analysis.

Once the Excel database was finalised, the network variables were calculated in the form of adjacency matrices, as outlined in the literature [48]. These adjacency matrices contained the 15 stamping facilities along the rows and columns of the matrix. For each of the network variables, the information in each cell of the corresponding adjacency matrix was populated as follows:

- Difference in time-to-adoption: absolute value difference in time-to-adoption in months between two stamping facilities
- Common scrap dealer: 1 if two stamping facilities used the same scrap dealer, 0 otherwise
- Faced resistance from scrap dealer: 1 if two stamping facilities both faced resistance from their scrap dealer, 0 otherwise
- Difference in size of scrap generated: absolute value difference in the size of scrap generated in tonnes between two stamping facilities
- Difference in order of implementation: absolute value difference in the order of implementation between two stamping facilities
- Significant capital investment: 1 if two stamping facilities both made significant capital investments, 0 otherwise
- Significant process changes: 1 if two stamping facilities both made significant process changes, 0 otherwise
- Difference in logistics costs: absolute value difference in logistics costs in GBP per tonne between two stamping facilities
- Common company owning facilities: 1 if two stamping facilities were both owned by the same company, 0 otherwise

The descriptive statistics for these network variables are summarised in Table 2.

The adjacency matrices for the dependent, independent, and control variables were then used to conduct the network regression using the Double Dekker Semi-Partialling MR-QAP algorithm in the software, UCINET [48]. The technique permutes multiple versions of the dependent variable adjacency matrix by randomly rearranging the data in the rows and columns. This creates independent variations of the dependent network variable with the same properties—mean, standard deviation, etc. Using this method, a sample of observations is generated for the network regression analysis. Since a larger sample of permutations provides more stable results, 10,000 permutations were specified in the UCINET software, following the example outlined in Borgatti, Everett, and Johnson [48]. Performing a

statistical analysis of these permutations enables us to see if the correlation between the variables is due to chance, or if there is a statistically significant correlation [53].

**Table 2.** Descriptive statistics for network regression variables.

| Variable | Min | Max | Mean | Std. Dev. |
|---|---|---|---|---|
| Dependent Variable | | | | |
| Difference in time-to-adoption | 0 | 18 | 7.68 | 5.55 |
| Independent Variables | | | | |
| Common scrap dealer | 0 | 1 | 0.30 | 0.46 |
| Faced resistance from scrap dealer | 0 | 1 | 0.27 | 0.44 |
| Control Variables | | | | |
| Difference in size of scrap generated | 0 | 1391 | 376.21 | 367.17 |
| Difference in order of implementation | 1 | 14 | 5.33 | 3.40 |
| Faced significant capital investment | 0 | 1 | 0.10 | 0.29 |
| Faced significant process changes | 0 | 1 | 0.20 | 0.40 |
| Difference in logistics costs | 0 | 68 | 19.92 | 15.62 |
| Common company owning facility | 0 | 1 | 0.11 | 0.32 |

**Note:** There were 210 observations.

## 4. Results

The results from the qualitative research follow the coding framework, which focused on three main relationships in the REALCAR network. The first is the coercive pressure from Jaguar Land Rover and how they used their network position and hierarchical relationships of authority and power as customers to encourage the stamping facilities to adopt REALCAR. The second is the mimetic pressure of peer stamping facilities, exploring how competitive and cooperative relationships, as well as interlocks between the stamping facilities, encouraged the learning and imitation of best practices and influenced the adoption of REALCAR. The third is the influence of scrap dealers, to determine if the changing structure and nature of these relationships had any effect on the adoption of REALCAR. Finally, the results from the network regression analysis to further explore the influence of the scrap dealer relationships are presented.

### 4.1. Coercive Pressures from Jaguar Land Rover on the Adoption of REALCAR

As described earlier, Jaguar Land Rover approached its suppliers to implement the REALCAR closed-loop approach across the supply chain and realise its benefits. Many interviewees mentioned the business benefit to Jaguar Land Rover, as the financial value and reduced environmental impact were big drivers for the implementation. However, there needed to be sufficient volumes and throughput to achieve these benefits. As one manager from Jaguar Land Rover stated, "it was within Jaguar Land Rover's interest to be able to roll this out extensively ... there is no point putting in massive conveyor belts and separation activities in a facility if you are only going to be separating a small amount of material."

To achieve a sufficient scale, stamping suppliers were the most important stakeholders to get on board. One interviewee stated that "it was in their interest to get their Tier 1 stampers to do it", and another mentioned that they needed to convince the stamping suppliers that REALCAR was important to the future strategy of Jaguar Land Rover. Thus, the focus of Jaguar Land Rover was to engage in discussions with their internal stamping facilities and external stamping suppliers to implement REALCAR.

Internal stamping facilities were "more straightforward since Jaguar Land Rover had a direct impact on them", as a manager from Jaguar Land Rover stated. Interviewees from Jaguar Land Rover and the internal facilities mentioned that necessary investments in equipment and workforce training to separate and collect the waste scrap material were made. There was little pushback against the

adoption of REALCAR according to the interviewees, since the approach made business sense and any investments in these facilities in terms of capital and process changes were very quickly paid back.

For the external stamping suppliers, a team of Jaguar Land Rover managers, external consultants, as well as the purchasing team from Jaguar Land Rover set up meetings and visits with all the external stamping supplier facilities, by order of size, to scale REALCAR quickly. Interviewees mentioned that a few external stamping suppliers were positive and understood the financial and environmental benefits of REALCAR, while others did not understand the closed-loop approach, and some did not want to do it. According to a manager at Jaguar Land Rover, many stamping suppliers were not focused on circular economy or sustainability and were more concerned about "getting press parts out at the right quality ... they weren't overly fussed about scrap."

Thus, to get their supply chain to adopt REALCAR, Jaguar Land Rover leveraged their position as customers to pressure suppliers that were dependent on them or that saw a potential for additional business, created requirements for future suppliers, and gave financial incentives to suppliers to ease the burden and the costs of implementing REALCAR. Figure 4 summarises the mentions of the various coercive pressures coded from the interviews.

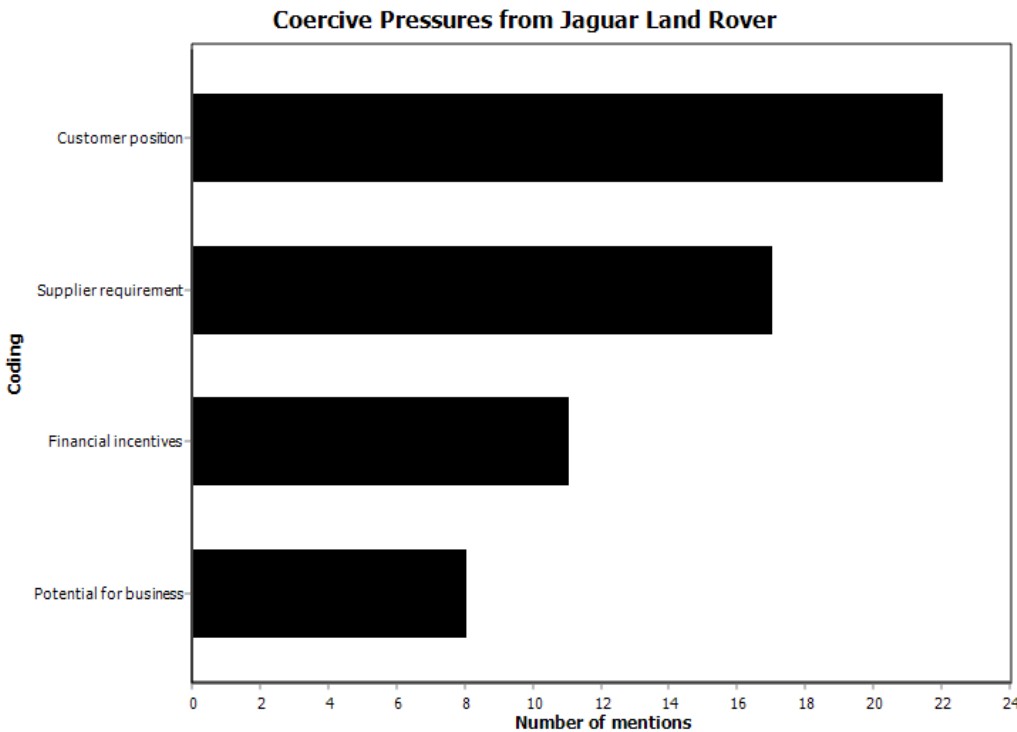

**Figure 4.** Coded mentions in interviews of coercive pressures from Jaguar Land Rover on suppliers to adopt REALCAR.

The majority of interviewees mentioned that Jaguar Land Rover is a big customer upon which external stamping suppliers were dependent, which compelled them to adopt REALCAR. They stated that in the UK, Jaguar Land Rover was the only mass volume automobile manufacturer using aluminium, so many companies were "strategic suppliers" which were exclusively stamping aluminium for them. Some external stamping suppliers had other customers that used steel, so while Jaguar Land Rover was not the majority of their overall business, it still represented a sizeable portion. As one interviewee stated, "when your biggest customer wants something, you try somehow to make it happen." So, there was "willingness to accommodate and adapt the processes" to implement the REALCAR closed-loop in order to "please Jaguar Land Rover" and maintain a positive relationship, according to a manager at an external stamping supplier.

Since Jaguar Land Rover was an important customer, they were able to request and direct their suppliers to adopt REALCAR. An executive at Jaguar Land Rover mentioned that the adoption of REALCAR became a "procurement rule for all pressing plants in the UK" and "a requirement for all their suppliers." There needed to be a commitment by the external stamping suppliers to participate in the REALCAR closed-loop and "there was never going to be a point where [suppliers] could say no since they could lose business", as another interviewee stated.

There was "no direct financial benefit" from REALCAR for the external stamping suppliers, since "they don't really make money out of it", according to an interviewee from an external stamping supplier. Thus, to encourage the adoption of REALCAR by external stamping suppliers, Jaguar Land Rover provided financial incentives to their suppliers. For some of the external stamping supplier facilities, there were minimal changes needed in their existing production processes to implement REALCAR, and Jaguar Land Rover would cover the costs for scrap separation. For other external stamping supplier facilities that needed more investment, some interviewees mentioned that "Jaguar Land Rover offered financial assistance for any CAPEX projects" that needed to be implemented to separate the scrap for REALCAR. Therefore, for many stamping suppliers, there was "no reason not to" implement the collection and separation of scrap for REALCAR, and it was a small price to pay to keep their business with Jaguar Land Rover.

Interviewees also mentioned that some external stamping suppliers saw a potential for business from the adoption of REALCAR. They recognised that Jaguar Land Rover was increasingly shifting towards recycled aluminium and understood that this was their new strategic direction. As a result, stamping suppliers and scrap dealers hoped that adopting REALCAR would enable them to grow their business and get further access to Jaguar Land Rover's supply chain. Others recognised the value of scrap and saw REALCAR as "a huge de-risking strategy when it comes to market movements", according to a manager from an external stamping company since they "didn't have to worry about negotiating prices and contracts to get rid of the aluminium waste."

Overall, it was apparent from the interviews that Jaguar Land Rover had a strong position as a customer in their supply chain. This enabled them to exert coercive pressures to influence their suppliers to adopt the REALCAR closed-loop approach. Suppliers had to adopt REALCAR to satisfy Jaguar Land Rover as a customer or risk losing their business relationship. However, this did not come at a huge cost to the stamping suppliers and even offered the possibility for additional future business.

*4.2. Mimetic Pressures among Peer Stamping Suppliers on the Adoption of REALCAR*

During the interviews, interviewees were asked about the nature of the relationships between the internal stamping facilities and external stamping supplier facilities. These questions were focused on identifying whether there were any competitive pressures to adopt REALCAR, any sharing of learning and best practices, and any employee interlocks between suppliers that enabled knowledge transfer. A summary of the coded mentions of these peer relationships is shown in Figure 5.

There were mentions of cooperative relationships and sharing of best practices during the interviews, but primarily among the three internal stamping facilities of Jaguar Land Rover. There was one internal facility that was highly advanced and first implemented REALCAR, and this facility exchanged knowledge and information to implement the same systems and processes in the other two internal facilities. As an operations manager from the internal facility stated, "I went over and did the trials . . . we used the same conveyor company . . . it helped with commonality between the three of us." Later on, as the other two internal facilities became more advanced, systems and processes were shared with the first, facilitating cross-learning.

When asked about the sharing of best practices among external stamping suppliers, interviewees mentioned that during the early stages of the implementation, Jaguar Land Rover hosted seminars and workshops to introduce REALCAR. During these events, managers from the internal stamping facilities and some external stamping supplier facilities which implemented the REALCAR approach were asked to "stand up and explain it to [their peers] and show [their peers] the process", according to a manager

from an external stamping facility. External stamping suppliers were also invited to visit the internal stamping facilities of Jaguar Land Rover to understand how REALCAR was implemented and apply those learnings. However, external stamping suppliers did not exchange knowledge or best practices among each other, since, as one interviewee from an external stamping supplier stated, "naturally, we can't have our competitors walk around our facilities, showing them our intellectual property."

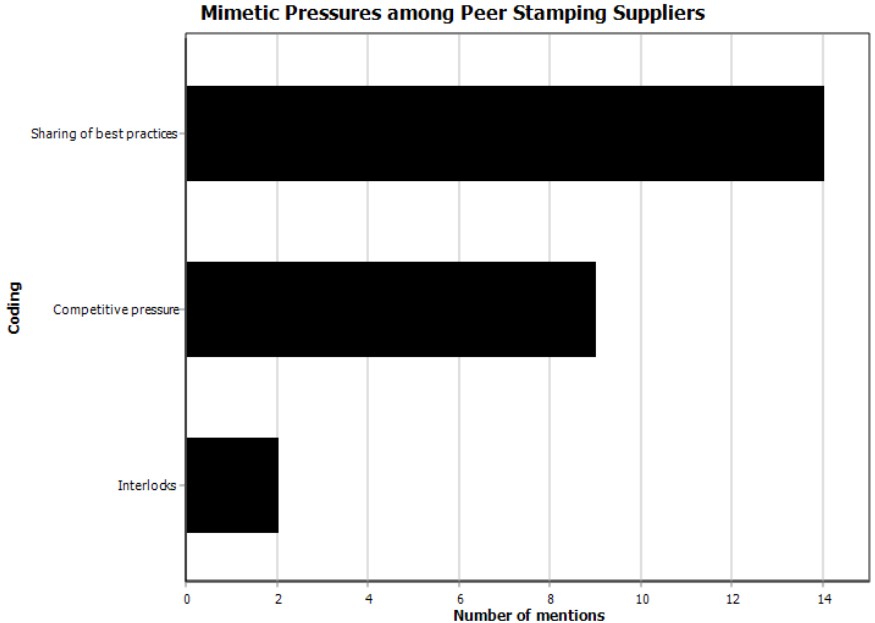

**Figure 5.** Coded mentions in interviews of mimetic pressures among peer stamping suppliers to adopt REALCAR.

As the implementation of REALCAR progressed, Jaguar Land Rover created a best-practice booklet based on the experiences of early adopters, which was updated as more and more stamping facilities implemented REALCAR. Consequently, as one manager described, "later [external stamping facilities] became easier because it was easier to tell them what to do." Another manager from Jaguar Land Rover who was involved in the discussions with the external stamping suppliers mentioned that it also became easier to convince suppliers to adopt REALCAR, likely due to the accumulated best practices from previous adopters.

Mentions of the influence of competitive pressures between stamping suppliers on the adoption of REALCAR were mixed. A few interviewees suggested that competitive pressures played some role in the decision to adopt REALCAR by suppliers. According to an interviewee from Jaguar Land Rover, competitive pressures were likely "much more important at the beginning of the project as it was very innovative and a new thing." One of the interviewees from an early adopter stated: "as a supplier partner, we always like to be at the forefront and lead not follow."

Other interviewees mentioned that there was no evidence of explicit competitive relationships. One manager stated that though external stamping companies were competitors, adopting REALCAR "because they say that if they did it, they would have a competitive advantage, I've never seen that." Interviewees described that the adoption of REALCAR was largely due to a fear of being cut out from the process by Jaguar Land Rover, suggesting a greater influence from Jaguar Land Rover rather than other competitors.

There were a few mentions during the interviews that engineers and suppliers talked to each other, hinting at the possibility of interlocking relationships. Interviewees mentioned that much of the REALCAR project was more bottom-up than top-down. This was confirmed through desk research on the 61 individuals reported in Table 1 to determine their roles at the time of REALCAR's implementation, from 2013 to 2017. The majority of individuals were managers, as shown in Figure 6.

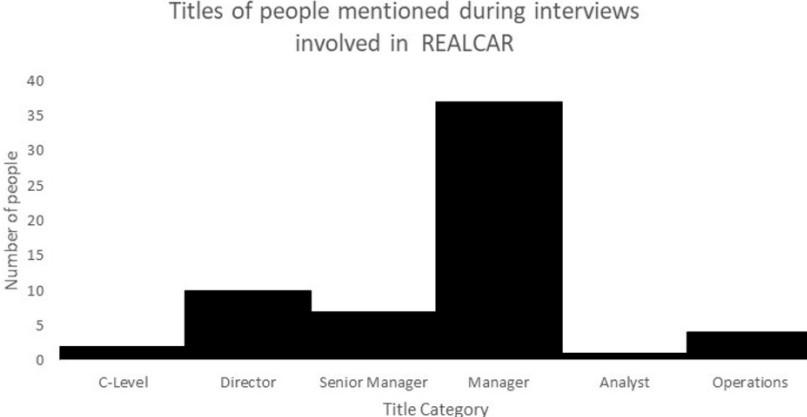

**Figure 6.** Roles of the 61 individuals identified in the research who were involved in REALCAR across the supply chain between 2013 and 2017.

Therefore, rather than identifying interlocking relationships at a board or executive level, which could indicate a flow of knowledge and information between firms, as with the analysis of Mizruchi [54], potential interlocks at the manager level were explored. Desk research using the public LinkedIn profiles of the 61 individuals was conducted to understand whether there was any movement from one stamping supplier to another during the timeframe of REALCAR from 2013 to 2017. However, the majority of individuals worked in the same company during the timeframe of REALCAR, and there was only one case of a manager moving from one external stamping supplier to another. This suggested that the transfer of information due to interlocks at the employee level likely did not play a role in the adoption of REALCAR.

Based on the results from the interviews, there was no clear evidence that competitive pressures or interlocks among the stamping suppliers influenced the decision to adopt REALCAR. While there was evidence of sharing of best practices among the suppliers, this was mainly facilitated by Jaguar Land Rover. They tried to encourage learning and knowledge sharing in the supply chain network by inviting external stamping suppliers to their internal facilities and asking them to share their experiences with REALCAR at seminars and workshops. Through the best-practices document that was started by Jaguar Land Rover and updated throughout the implementation, it became easier for later stamping facilities to learn from, and implement, the best practices of previous adopters.

*4.3. Changing Structure and Nature of Scrap Dealer Relationships on the Adoption of REALCAR*

Of the various ties in the Jaguar Land Rover supply chain network, the relationships with the scrap dealers that handled the waste material were the most affected by the REALCAR project. The dynamics of these relationships between Jaguar Land Rover, the stamping facilities, and the scrap dealers placed various pressures on the adoption decisions of REALCAR. Figure 7 summarises the coded mentions during the interviews regarding the changing structure and nature of scrap dealer relationships during REALCAR.

Many of the external stamping suppliers and scrap dealers had long-term relationships and existing contracts that were challenged by the new REALCAR approach. "Some of them had in the contract that the scrap belongs to them . . . so the materials and scrap generated was part of the service fee . . . this was a hurdle that we had to go step-by-step . . . to see how to change those contracts", as one interviewee stated. Jaguar Land Rover had to identify the scrap dealers contracted to their suppliers and have separate meetings with them to determine "how to take over the scrap stream", according to a manager at Jaguar Land Rover. According to another manager, some of these meetings became quite heated and "we were escorted out of the premises."

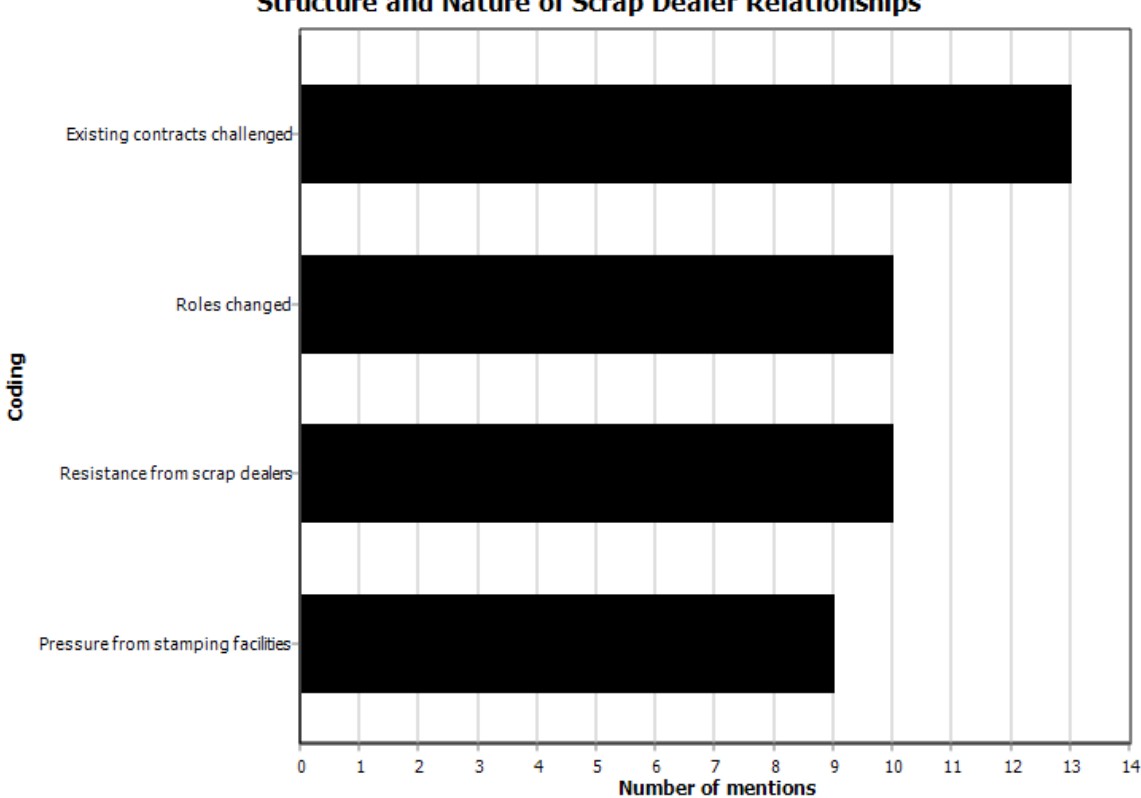

**Figure 7.** Coded mentions in the interviews regarding the changing structure and nature of the relationships with scrap dealers.

This was because the role of the scrap dealer changed in REALCAR's closed-loop model. As one interviewee mentioned, REALCAR altered the "ownership and power control between the people selling the aluminium sheet and the people selling or buying back the scrap." Figure 8 illustrates how the supply chain relationships differed under REALCAR, particularly for the scrap dealers.

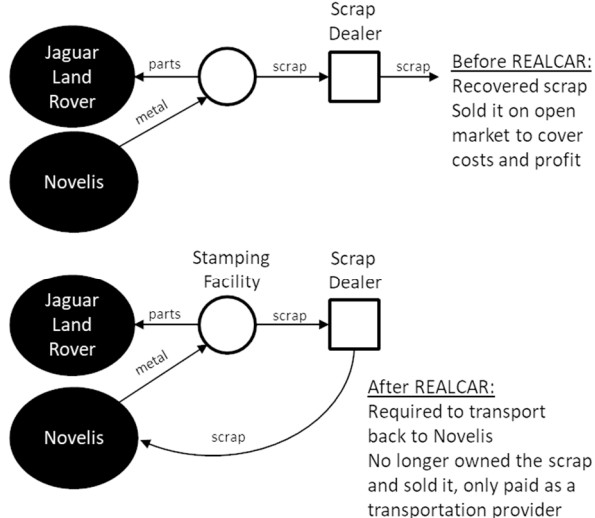

**Figure 8.** Diagram of the changing nature of the relationships and the role of scrap dealers in the supply chain under REALCAR.

Before REALCAR, scrap dealers made a business of providing equipment to stamping facilities to collect the scrap and then selling it on the open market to cover their costs and make a profit.

Under REALCAR, they were asked to change their business model to get paid for the transportation of the aluminium waste, what they called "running wheels." As one interviewee described, "originally they would have maybe bought the metal for 900 a ton and sold it for 1000 and made a profit of 100, but now maybe they would get 14 a ton." Another interviewee said the scrap dealers, particularly the smaller ones, probably "don't know the true costs" of their operations and likely were not sure "if they are actually making money or not" with their new role in the closed-loop process.

Moreover, Jaguar Land Rover, Novelis, and the external stamping suppliers had different preferred scrap dealers they wanted to work with. According to one interviewee from Jaguar Land Rover, this created "frosty relationships" among certain scrap dealers and resulted in difficulties in negotiating the adoption of REALCAR. A few of the larger external stamping suppliers that had more leverage tried to make compromises and negotiate agreements with Jaguar Land Rover and Novelis to maintain relationships with their preferred scrap dealers.

A few scrap dealers understood the benefits of REALCAR quite quickly and, according to an interviewee involved in discussions, "realised that in the future the ultimate goal is recycling" and that "they could get introduced to new business this way as well." However, some of the smaller, more local scrap dealers tried to hinder the adoption of REALCAR by paying more for the scrap from the external stamping suppliers or committing to buying scrap months in advance. Thus, a few of the external stamping suppliers which had agreed to adopt REALCAR "started dragging their feet", according to this interviewee, and "became obstructive and delayed the process" for the adoption of REALCAR.

As a result, the relationships between Jaguar Land Rover, the external stamping suppliers, and the scrap dealers became strained due to REALCAR. Multiple interviewees mentioned that for the external stamping suppliers, the relationship with Jaguar Land Rover was more important than the relationship with the scrap dealers, and as one interviewee from Jaguar Land Rover stated, "it was a touchy subject." An interviewee from a scrap dealer mentioned that REALCAR "was always going to happen, and it made a lot of sense because it was a direction coming from Jaguar Land Rover . . . it was inevitable."

Just as Jaguar Land Rover put pressure on external stamping suppliers to adopt REALCAR or lose the contract, external stamping suppliers put pressure on the scrap dealers. One interviewee from an external stamping supplier said they told their scrap dealer that "this was the way forward and they should embrace it otherwise they could lose the contract." In some cases, stamping facilities had to change their scrap dealers to adopt REALCAR without any problems. Ultimately, scrap dealers relented to the adoption of REALCAR, since "they realised they needed to get on board or they would get nothing", as one manager from an external stamping supplier stated.

From the interviews, it was clear that the structure and nature of the relationships with the scrap dealers in Jaguar Land Rover's supply chain network were changed due to REALCAR. Scrap dealers had to alter their business model and role in the network, which caused them to push back and strain existing relationships with the external stamping suppliers and Jaguar Land Rover. External stamping suppliers placed more importance on the relationship with Jaguar Land Rover than their scrap dealers, so when they were pressed to adopt REALCAR, this cascaded to the scrap dealer as well.

*4.4. Results of the MR-QAP Network Regression*

The changing structure and nature of the relationships of the scrap dealers was a particularly interesting outcome from the qualitative research. On the one hand, scrap dealers seemed to resist and tried to delay the adoption of the closed-loop approach, but on the other hand, they faced pressure from Jaguar Land Rover and the stamping suppliers to accept their role in the REALCAR model and keep their business. This was further explored through the MR-QAP network regression analysis to test how significant a role the relationships with scrap dealers played in influencing the adoption timeframes of REALCAR.

The results of the MR-QAP regression are shown in Table 3. The model's R-squared is 0.35, and the adjusted R-squared is 0.32, suggesting that there are more variables that we have not measured which could be influencing the dependent variable [48]. However, since 32% of the observed variation can be

explained by the variables included and the P(r2) is highly significant, the results of the model are worth exploring to complement the qualitative analysis and gain a better understanding of the role of the scrap dealer relationships.

**Table 3.** MR-QAP network regression on the dependent variable, difference in time-to-adoption.

| Variable | Std. Coeff. | *p*-Value |
|---|---|---|
| Independent Variables | | |
| Common scrap dealer | −0.21 | 0.02 ** |
| Faced resistance from scrap dealer | −0.43 | 0.002 *** |
| Control Variables | | |
| Difference in size of scrap generated | −0.12 | 0.08 * |
| Difference in order of implementation | 0.19 | 0.04 ** |
| Faced significant capital investment | 0.08 | 0.13 |
| Faced significant process changes | −0.03 | 0.31 |
| Difference in logistics costs | 0.10 | 0.12 |
| Common company owning facility | −0.08 | 0.19 |

Notes: There were 210 observations. The analysis was done with 10,000 permutations. The R-Squared is 0.35, the adjusted R-squared is 0.32, and P(r2) is 0.001. *, **, *** indicate significance at the 10%, 5%, and 1% levels, respectively.

As the results show, scrap dealer resistance has the largest influence on time-to-adoption and is the most statistically significant variable, with a 99% confidence level. The high, negative standardised coefficient means that any two stamping facilities facing resistance from their scrap dealers had similar adoption timeframes to implement REALCAR, controlling for other factors. This provides additional quantitative evidence for what emerged during the qualitative interviews, suggesting that resistance from scrap dealers did play an important role in how long it took for the stamping facilities to adopt REALCAR.

In addition, the common scrap dealer variable was also significant at a 95% confidence level. It also had a high, negative standardised coefficient, suggesting that any two stamping facilities that shared a common scrap dealer were more likely to have similar time-to-adoption of the REALCAR closed-loop approach, controlling for other factors. Thus, in addition to explicit resistance from the scrap dealers, it seems that other aspects of the relationship with the scrap dealers and their reactions to REALCAR also played a role in influencing the adoption timeframes of the stamping facilities to implement the closed-loop approach.

Among the control variables, the difference in the order of implementation had a positive and statistically significant effect, with a 95% confidence level. The positive standardised coefficient showed that stamping facilities that are further apart in terms of the order of implementation were more dissimilar in terms of time-to-adoption, perhaps suggesting an effect of firm learning. Although the potential size of scrap had a negative relationship with time-to-adoption, it was only statistically significant at a 90% confidence level. The remaining control variables did not seem to have a statistically significant relationship with the difference in time-to-adoption between stamping facilities.

## 5. Discussion and Conclusions

The qualitative interviews and the network analysis of the REALCAR case study describe the importance of embedded relationships in the adoption of circular economy eco-innovations such as closed-loop recycling. Compared to traditional innovations, the novel processes and investments required for the REALCAR eco-innovation, the uncertainty of the economic benefits for Jaguar Land Rover's suppliers, and the changing nature of the relationships within the supply chain meant that there were differences in incentives and motivations for the adoption of REALCAR among Jaguar Land Rover and their suppliers. As a result, the coercive pressures by Jaguar Land Rover to encourage suppliers to adopt REALCAR, their facilitation of the exchange of best practices, and the resistance

from scrap dealers to their new role played a significant role in influencing the timeframes for the adoption of REALCAR.

For Jaguar Land Rover, the decision to implement the REALCAR eco-innovation had clear financial and environmental benefits. Using recycled aluminium from the closed-loop process enabled them to achieve their goals of creating more lightweight and fuel-efficient vehicles, reducing environmental emissions, and lowering production costs [41]. However, as described in the literature [30,32], novel eco-innovations like REALCAR, which affect the whole supply chain, required the intense cooperation of Jaguar Land Rover's suppliers. Therefore, it was in Jaguar Land Rover's interest to convince as many of their suppliers as quickly as possible to adopt the closed-loop approach to fully realise its benefits.

The suppliers of Jaguar Land Rover, on the other hand, did not have the same motivations. While some of the stamping facilities understood the environmental benefits, the majority had low financial and environmental drivers to adopt REALCAR, as it was not a part of their core business. To overcome this, the nature of the relationships between Jaguar Land Rover and its suppliers was especially important. The research by Williamson [38] shows that SMEs will not voluntarily adopt eco-innovations unless it satisfies criteria for business performance such as satisfying their customers' needs, because many companies do not perceive a clear benefit in being environmentally responsible [31]. Furthermore, a study on automotive suppliers describes how customer requirements are one of the major factors in their participation in green initiatives, as well as, to a certain extent, cooperative supplier relationships and investments from their customers [55]. As such, Jaguar Land Rover's pressure as a customer, their creation of supplier requirements, and their provision of financial incentives were all needed to offset the costs of capital investments and process changes to incentivise their suppliers to adopt REALCAR.

Moreover, while traditional innovation and eco-innovation literature describes competitive pressures to adopt innovation and the use of networks to learn from, and imitate, the best practices of peers and competitors [5,21,24–26,30], there was no clear evidence of this happening in the case of REALCAR. Rather than any competitive pressures, the motivation of suppliers to adopt was primarily the fear of being cut out of the relationship with Jaguar Land Rover. External stamping facilities had no interest in exchanging best practices, relying instead on learning from Jaguar Land Rover's internal stamping facilities and utilising their best practices document. This is likely also due to the low financial and environmental motivations for the majority of the stamping facilities to adopt the REALCAR eco-innovation. Jaguar Land Rover had to leverage their position and relationship as customers and act as a hub firm [22] to orchestrate the exchange of knowledge among internal and external stamping facilities.

Lastly, scrap dealers were the most affected by the REALCAR closed-loop approach. The new process changed the structure and nature of the relationships in Jaguar Land Rover's supply chain and altered the scrap dealers' role and their business model. While a few of the scrap dealers recognised this shift and accepted REALCAR, many scrap dealers were against it and even tried to hinder the adoption of the closed-loop approach by trying to entice stamping facilities with better terms. The results of the network regression showed that this resistance from the scrap dealers had a significant effect on the adoption timeframes of the stamping facilities, controlling for other factors. In the end, it took heated discussions as well as coercive pressures from Jaguar Land Rover and the stamping facilities before the scrap dealers had to either accept their new role or lose their contracts.

The results from this case study show how the embedded relationships and ties between firms in supply chain networks that influence innovation adoption are likely to play an even stronger role in eco-innovation adoption. Circular economy eco-innovations like closed-loop recycling require the cooperation and alignment of the economic and environmental goals of the entire supply chain [3], particularly the close partnership and collaboration of suppliers [56,57] and their customers [58]. However, the goals of the various stakeholders may not be aligned, as was the case with REALCAR. While there were a few suppliers that understood the financial and environmental benefits of REALCAR, many did not have these motivations to adopt and largely complied to keep their business, and some

even acted to hinder the adoption. In this case, companies like Jaguar Land Rover that wish to push the adoption of an eco-innovation across their supply chain need to take into account the embedded relationships and the structure and nature of their supplier networks. By understanding their position and the strength of their relationships within their supply chain network and the changing dynamics of inter-firm network relationships, companies can exploit favourable conditions, improve conducive factors, or remove obstacles to accelerate the adoption of circular economy eco-innovations.

Although this paper's exploration of the single case study of REALCAR cannot be generalised to all eco-innovations, nor was that the intent of this research, it does provide an example of a particular circular economy eco-innovation, i.e., closed-loop recycling. Further research is warranted to determine whether such embedded relationships in inter-firm networks play a role in other types of circular economy eco-innovations. As in the case of REALCAR, coercive pressures from power dynamics in customer relationships, mimetic pressures from peer learning, and the changing structure and nature of relations with suppliers in inter-firm networks could influence the adoption and time-to-adoption of eco-innovations in other cases. Perhaps additional structural and relational embedded relationships reported in the literature, such as hierarchical relations of authority and power, competitive or cooperative relationships, interpersonal ties and interlocking directorates, or joint membership in associations [5,6,16,18] could impose various multiplex pressures [19,20] on circular economy eco-innovation adoption.

**Funding:** The International Aluminium Institute provided financial assistance to attend aluminium industry conferences to further develop this study.

**Acknowledgments:** The author would like to thank the interviewees for contributing to this research with their valuable time and insights. He would also like to thank Flaminio Squazzoni, Federico Bianchi, and Niccolò Casnici for their recommendations and feedback on the research design of this study.

**Conflicts of Interest:** The authors declare no conflict of interest. The funders had no role in the design of the study; in the collection, analyses, or interpretation of data; in the writing of the manuscript, or in the decision to publish the results.

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
