# Peer review of "Influence of Inter-Firm Network Relationships on Circular Economy Eco-Innovation Adoption"

_sustainability, doi:10.3390/su12187607_

Round 1

Reviewer 1 Report

The modifications made by the authors answer the requests I made, thanks to them

Author Response

Dear Reviewer,

Thank you very much for your feedback, and for your comments on the earlier version of the manuscript.

Kind Regards,

Shyaam Ramkumar

Reviewer 2 Report

I can say that the authors have dealt with my comments on the first version in an absolutely satisfactory way and have integrated all the suggestions I have proposed.

Author Response

Dear Reviewer,

Thank you very much for your comments on the earlier version of the manuscript. In the latest revision, the manuscript was revised to correct for spelling and grammatical errors and have more concise language to accommodate your comments requiring moderate English changes to the text.

Kind regards,

Shyaam Ramkumar

Reviewer 3 Report

I stick to my old opinion

Author Response

Dear Reviewer,

Thank you for your feedback. Based on other reviewer’s feedback regarding English changes in the text and minor spell-check, the latest revision of the manuscript has incorporated these changes in the hopes that it also clarifies the description of the methods, results, and conclusions as pointed out in your review. To further address your comments and suggestions, lines 344 to 352 of the manuscript reference literature using network regression analysis as a complement to qualitative analysis, and additional text was added in lines 355 to 356 and lines 705 to 706 to reiterate the complementary nature of the network regression analysis in this study.

Kind regards,

Shyaam Ramkumar

Reviewer 4 Report

Taking into account the revisions and the answer provided by the authors to my comments, I believe that the manuscript has been improved and now warrants publication in Sustainability.

Author Response

Dear Reviewer,

Thank you very much for your comments on the earlier version of the manuscript. In the latest revision, the manuscript was revised to correct for spelling and grammatical errors and have more concise language to accommodate your comments requiring minor spell check and other reviewers' feedback requiring moderate English changes to the text.

Kind regards,

Shyaam Ramkumar

This manuscript is a resubmission of an earlier submission. The following is a list of the peer review reports and author responses from that submission.

Round 1

Reviewer 1 Report

Thank you very much for the opportunity to review the manuscript entitled „Influence of Inter-Firm Network Relationships on Circular Economy Eco-Innovation Adoption “. The article is overall well-written, and it is of interest for the readership of Sustainability journals as it discusses the relation between inter-firm network relationships and eco-innovation.

I would like to encourage authors to consider several issues to be improved. I believe that after incorporating these issues, your paper will have a value for this journal. I hope that my comments are useful for authors, as they further develop the manuscript.

First, I think that the authors should improve the section of the literature review with more references regarding the innovation system approach. Some example are:

Edquist, C., & McKelvey, M. (2000). Systems of innovation: growth. In Competitiveness and employment. Cheltenham: Edward Elgar Publishers.

Malerba, F., & Mckelvey, M. (2018). Knowledge-intensive innovative entrepreneurship integrating Schumpeter, evolutionary economics, and innovation systems. Small Business Economics. 1-20.10.1007/s11187-018-0060-2.

Also, evolutionary economics theory regarding the existence of co-evolutionary processes in the economy in relation to knowledge and innovation is relevant for the topic of this article. Please see:

Murmann, J. (2013). The coevolution of industries and important features of their environments. Organization Science, 24(1), 58–78.

Second, I think that the paper should be re-organized. The methodology section should also include the description of the network analysis and more details that are placed in the appendix need be included in the article, such as the unit of analysis (nodes of the network) and how the considered variables have been constructed. Also, I suggest to change the name of the variable “firm size” as it usually refers to the number of employees. I am not sure about the relevance of “firm learning” as control variable, considering the dependent variable. Also, maybe you are able to include among the control variables the turnover of the company which owned the stamping facility. One main predictor for the adoption of circular economy practices is the turnover of the companies, according to:

Zamfir, A.-M.; Mocanu, C.; Grigorescu, A. Circular Economy and Decision Models among European SMEs. Sustainability 2017, 9, 1507.

Finally, I think that it would be interesting for the author to calculate and discuss some descriptive indicators for the network map of the dependent variable.

Reviewer 2 Report

The research design is too simple, only using regression method to complete the conclusion derivation. Data reliability and validity test are missing, unable to verify the reliability of data source.

Reviewer 3 Report

The article entitled “Influence of Inter-Firm Network Relationships on Circular Economy Eco-Innovation Adoption” deals with the adoption of eco-innovation in industrial network.

The theme is interesting and falls within the topics of the journal.

The article is well structured and well written, hovever, I believe it needs some minor improvements before being published.

Introduction:

  • The research gap is well presented.

Literature review:

  • Lit review is accurate, but why authors decided to directly start with subheaders?

Methods:

  • Row 192 “…have “no good” measure..” Is there a more accurate formulation? The intention is probably to say these concepts are not measurable quantitatively and thus need to be approached from a qualitative protective (not no good, but different).
  • For coding, could the exact meaning of “power and coercive pressures, competitive or cooperative relationships and mimetic pressures”, be better defined either here or in the literature review?

Results:

  • Figure 2 needs to be larger, font almost unreadable
  • Overall, this section is a bit too long and at some points repetitive. Only use quotes when absolutely necessary and make sure the content is not duplicated in other quotes.
  • In this section, the codes introduced in the methods section are hardly mentioned. What was the reason for their introduction, if they are not further elaborated on in the result section?
  • Currently, it feels more like a story or report than an analysis. Maybe mentioning the codes in the three parts of the results section would be helpful and give it a more analytical lens which was introduced from row 296-301?

Further exploration of scrap dealer relationship

  • Introduction can be shortened, at points repetitive
  • Figure 5 not necessary or should be altered, it does not contribute to understanding
  • The whole subchapter 5.1 should be part of the methods section. As the case study is already explained in detail in the methods section, the reason on why to do a network analysis can already be anticipated there. It does not fit in the 5thchapter, which should only contain results or discussions. As mentioned in the introduction of this paper, you have chosen to do a mixed methods approach which should all be explained at once.

Discussion:

  • Contribution to research on eco-innovation adoption is mentioned but not well explained. What is the difference between the inter-firm relationship influence on normal innovation vs. eco-innovation? g. that the whole supply chain needs to be taken into account to understand the systemic impact of lagging scrap dealers on the performance of the closed-loop as a whole. This is specific to closed loops, and might not be the case in normal innovations (maybe also not necessarily eco-innovations, which are not closed loop…)
  • Different kinds of pressure mentioned in this section but not in result section… It might be worth to show these pressures graphically for all the three relationships described in the result section. This could either be put in the result or the discussion section.

Appendix:

  • Are the figures needed without description of the nodes? Would either make figures more clear in what they represent or leave them out

References:

  • Look ok, just add doi to all sources (where available, e.g. missing for reference 1, 10, 11)

Style and typos:

There is a word repetition (followed) in row 76

Row 465-467 is a repetition.

Don’t use “And” at the beginning of a sentence e.g. row 199, 175, 405

Row. 197 is “conceived” the right word?

Row. 216, British pounds

Writing style could be less casual, avoid words like good, easy, happy etc

Reviewer 4 Report

The article proposes to deal with the role of inter-firm relations on the diffusion of eco-innovations. In particular, it proposes a review of the interesting literature on the subjects of the analysis of these relations. However, it is surprising not to see the founding works of WIlliamson and the authors who have continued the analysis of his results. With regard to the theme of eco-innovation, it would have been expected to go beyond the definition of what "eco-innovation" is by convincingly outlining how the study of its diffusion raises questions of renewed research with regard to its adoption by the value chain. Several are suggested in the text of the literature review (radicality of innovation, non-obvious economic gain...) but not debated in the discussion to highlight the specificities of the case of eco-innovation.

The work produced to present this article is substantial and the author has taken great care to set out his research methodology and to justify it in a meaningful way. However, it would be important to indicate the duration of the interviews conducted, whether they were face-to-face or by other means. Whether or not the coding was duplicated. With regard to the network analysis, no justification is proposed for the choice of variables, although subsequent analysis shows that they play only a partial role: 32% of the variation expressed. Part of Appendix A, which describes the networks, does not seem necessary for this article because their consultation is of little value, even though the author has been asked to do some work on their constitution. It was a necessary step in his work but it is not informative for the reader.

The major difficulty of the article is that it concludes that the situation of eco-innovation adoption in a network of actors belonging to a supply chain is not different from situations where the innovations to be disseminated are not eco-innovations. The situation studied turns out to be a situation of adoption of new technologies and transformation by a socio-technical system as it arises whatever the nature of the innovation to be adopted by such a system. This raises the initial question of the specificities of the adoption of eco-innovations. The specificities of eco-innovations, which could differentiate them from other cases of disruptive innovation imposed on existing socio-technical systems, are not raised and therefore not dealt with in the following analysis. As a consequence, the study loses interest in the creation of knowledge related to the context of eco-innovations. This raises the question of the added value of the publication in relation to the numerous studies conducted on the relations between the actors of the value chains, in particular as an extension of those of Williamson.

In conclusion, we recommend to rework the article which we think has a good potential since the author can better highlight the specificities of eco-innovation in the results of his study and integrate its aspects in the final discussion of his article. If this is not possible, the positioning of this article in the journal "Sustainability" does not seem easy to justify.